# A Simulation Environment and Reinforcement Learning Method for Waste Reduction

**Sami Jullien**                                                                 *s.jullien@uva.nl*
*AIRLab*
*University of Amsterdam*
*Amsterdam, The Netherlands*

**Mozhdeh Ariannezhad**                                          *m.ariannezhad@uva.nl*
*AIRLab*
*University of Amsterdam*
*Amsterdam, The Netherlands*

**Paul Groth**                                                                  *p.groth@uva.nl*
*University of Amsterdam*
*Amsterdam, The Netherlands*

**Maarten de Rijke**                                                       *m.derijke@uva.nl*
*University of Amsterdam*
*Amsterdam, The Netherlands*

**Reviewed on OpenReview:** *https://openreview.net/forum?id=KSvr8A62MD*

## Abstract

In retail (e.g., grocery stores, apparel shops, online retailers), inventory managers have to balance short-term risk (no items to sell) with long-term-risk (over ordering leading to product waste). This balancing task is made especially hard due to the lack of information about future customer purchases. In this paper, we study the problem of restocking a grocery store's inventory with perishable items over time, from a distributional point of view. The objective is to maximize sales while minimizing waste, with uncertainty about the actual consumption by costumers. This problem is of a high relevance today, given the growing demand for food and the impact of food waste on the environment, the economy, and purchasing power. We frame inventory restocking as a new reinforcement learning task that exhibits stochastic behavior conditioned on the agent's actions, making the environment partially observable. We make two main contributions. First, we introduce a new reinforcement learning environment, *RetaiL*, based on real grocery store data and expert knowledge. This environment is highly stochastic, and presents a unique challenge for reinforcement learning practitioners. We show that uncertainty about the future behavior of the environment is not handled well by classical supply chain algorithms, and that distributional approaches are a good way to account for the uncertainty. Second, we introduce GTDQN, a distributional reinforcement learning algorithm that learns a generalized Tukey Lambda distribution over the reward space. GTDQN provides a strong baseline for our environment. It outperforms other distributional reinforcement learning approaches in this partially observable setting, in both overall reward and reduction of generated waste.

# 1 Introduction

Retail is an industry that people deal with almost every day. Whether it is to sell clothes, groceries, or shop on the internet, all retailers require optimized inventory management. Inventory management considers a multitude of factors. One of growing concern is *waste*. For example, food waste costs the worldwide economy around \$1 trillion per year.[1] On top of this cost, food waste is responsible for around 10% of worldwide carbon emissions.[2] This is one order of magnitude higher than civil aviation.[3] This means that both businesses and non-profits have an interest in working together to reduce waste, given its economic and ecological impact.

While some waste is produced during production, retailers and consumers also play a significant role in the generation of food waste. In this paper, we focus on the retailer-side of waste. We use grocery stores as a canonical example of a retailer. Grocery stores need to manage their inventory in order to meet customer demand. To do so, they pass orders to warehouses. When restocking an inventory, an order is made to receive $n$ units of a product at a later time. Often, stocks are provisioned in order to ensure customers always have access to an item (Horoś & Ruppenthal, 2021). This means that, in the case of perishable items, they might waste items that have stayed in stock for too long. On the other hand, if items are under-stocked, it might lead to customers not finding the products they want. This results in a balancing problem where orders have to account for uncertainty in demand, both to minimize waste and meet customer demand. This process is of course repeated over several periods – a grocery store is usually open 6 to 7 days a week. This makes inventory replenishment a sequential decision making problem, where actions have potentially delayed outcomes.

**Simulation environment.** Simulation environments have proven promising for supply chain problems (Cestero et al., 2022), as they allow for experimentation from both the concerned community and machine learning experts. Currently, there is no available framework that allows us to properly simulate a grocery store that takes waste into account for different items. Hence, to help evaluate the performance of agents on the inventory restocking (or inventory replenishment) problem, we introduce a grocery store environment that takes waste and stochastic customer demand into account.

**Learning method.** The stochasticity of customer consumption makes the inventory replenishment problem *partially observable*: the demand being different from its forecast, two identical situations at first sight can result in different outcomes. This creates a problem: if an action, for a given observation, can result in various rewards, how do we ensure that we properly learn the dynamics of the environment? A possibility is to consider non-deterministic action-value functions, where we ascribe the randomness in the environment to its reward distribution. Given this, as a strong baseline, we propose to make use of distributional reinforcement learning (DRL). In DRL, the agent aims to estimate the *distribution* of the state-action value function $Q$ rather than its *expectation* (Bellemare et al., 2017). In this paper, we adopt a new direction to estimate the distribution. Non-parametric estimations of summary statistics of the probability distribution are preferred for unconventional data distributions, but are often prone to overfitting and require more samples (Pados & Papantoni-Kazakos, 1994; Sarle, 1995). To circumvent this limitation, we estimate parameters of a flexible distribution, in order to facilitate learning. Actual reinforcement-learning based approaches to waste reduction in the inventory problem do not look at the item-level (Kara & Dogan, 2018). We aim to fill this gap in perishable item replenishment by making use of distributional reinforcement learning.

We introduce GTDQN, *generalized Tukey deep Q-network*, a reinforcement learning algorithm that estimates parameters of a well-defined parametric distribution. Currently, distributional approaches rely mostly on non-parametric estimation of quantiles. We find that distributional algorithms with a reliable mean estimate outperform non-distributional approaches, with GTDQN outperforming expectile-based approaches. While we focus on the task of inventory replenishment, GTDQN does not make any assumption on the task we present here.

---

[1] World Food Programme, `https://www.wfp.org/stories/5-facts-about-food-waste-and-hunger`

[2] WWF: Driven to Waste, `https://wwf.panda.org/discover/our_focus/food_practice/food_loss_and_waste/driven_to_waste_global_food_loss_on_farms`

[3] `https://www.iea.org/reports/aviation`

**Research questions.**   Overall, we aim to answer the following research questions:

1. Given a forecast for the consumption of a perishable item, can we find an optimal strategy to restock it while maximizing overall profits?

2. Can we ensure that such a policy does not lead to increased waste?

3. Which distributional method is the most efficient to solve the problem?

To answer those questions, we compare various discrete-action based DRL methods, including our newly proposed GTDQN, as well as classic inventory replenishment heuristics. Previous work has tried to answer those questions partially. Meisheri et al. (2022) do not look at waste through a cost-based approach. And De Moor et al. (2022); Ahmadi et al. (2022) solely look at a single item, with unchanging demand distribution. Likewise, Selukar et al. (2022) look only at a very limited number of items and only consider the problem in a LIFO manner. Overall, the previous work does not provide a common solution for practitionners to try new ordering policies, nor do they provide new ordering algorithms.

**Contributions.**   In summary, our contributions are as follows:

- We provide *RetaiL*, a new, complete simulation environment for reinforcement learning and other replenishment policies based on realistic data;

- We showcase the performance of classic reinforcement learning algorithms on *RetaiL*;

- Additionally, we propose GTDQN, a new distributional reinforcement learning algorithm for the evaluation of state-action values; and

- We show that GTDQN outperforms the current state-of-the-art in stochastic environments, while still reducing wastage of products, making it a strong baseline for *RetaiL*.

Below, we survey related work, introduce our simulation environment, discuss baselines for sales improvement and waste reduction in this environment, including our newly proposed distributional reinforcement learning method, report on the experimental results, and conclude.

## 2   Related Work

**The inventory restocking problem**   The literature on ordering policies is extensive. Most work is based on the classic $(s, S)$ policy introduced by Arrow et al. (1951). Yet, their inventory model does not factor in waste. Inventory policies for fresh products as a field was kick-started to optimize blood bag management (Jennings, 1968; Brodheim et al., 1975). Since then, there is increased attention in the classic supply chain literature models to limit waste (van Donselaar et al., 2006; Broekmeulen & van Donselaar, 2009; Minner & Transchel, 2010; Chen et al., 2014). Recently, various reinforcement learning-based policies have been developed for supply chains; see (e.g., Kim et al., 2005; Valluri et al., 2009; Sui et al., 2010; Gijsbrechts et al., 2019; Sun et al., 2019). More specifically, Kara & Dogan (2018) pioneered the use of reinforcement learning for waste reduction in the inventory restocking problem by using a DQN to solve the problem at hand. Their approach can be improved upon, as they aggregate the total shelf lives of the items at hand – thus, their agents only have access to the average shelf life of the inventory. Moreover, this makes it impossible to account for all items independently, to remove expired items from the stock, and to penalize the agent for the generated waste. Indeed, waste can be considered a tail event as it happens suddenly once an item has reached its maximum consumption date. Item-level waste is currently not considered in the literature. This is why we advocate for simulations where the agent considers all of the inventory.

We think it is not enough to limit the agent's knowledge by only looking at the mean. Indeed, a distribution has more summary statistics than its first moment, especially to characterize its tail. We should make use of those, and we believe that this is required for a proper evaluation of waste. With distributional reinforcement learning, the agent can learn its own summary characteristics, which will be more suited to the task at hand.

**Partially observable Markov decision processes.** Randomness in environments is common in reinforcement learning (Monahan, 1982; Ragi & Chong, 2013; Goindani & Neville, 2020). We can distinguish two approaches to this stochasticity, that are not necessarily disjoint. The first is to consider robust Markov decision processes. They make the assumption that a policy should be robust to changes in the data generating process over time, in order to have a better estimation of the transition matrix (Xu et al., 2021; Derman et al., 2020). The other approach is to consider the reward as a non-deterministic random variable whose distribution is conditioned on the environment's observation and on the agent's action. This usually means that the agent acts under partial information about the environment's state. While one can make the argument that this is only due to the lack of information about the environment (Doshi-Velez, 2009), this is not a setting that generalizes well to unseen situations.

In this paper, we consider that our agent can see the current state of the stock for a given item and its characteristics, but lacks the information over the past realizations of the temporally joint demand distribution, making the environment partially observable.

**Distributional reinforcement learning.** Learning the $Q$-value is the most straightforward way to develop a $Q$-learning algorithm, but is most likely to be inefficient, as noted by Bellemare et al. (2017). Bellemare et al. introduce the C51 algorithm, where they divide the possible $Q$-value interval in 51 sub-intervals, and perform classification on those. The goal is to learn the distribution of future returns instead of their expectation $Q$. This allows one to achieve a gain in performance, compared to using only the expectation; this paper launched the idea of distributional deep reinforcement learning. Later, the authors introduced a more generalizable version of their algorithm, the quantile-regression DQN (Dabney et al., 2018). Instead of performing classification on sub-intervals, Dabney et al. directly learn the quantiles of the $Q$-value distribution through the use of a pinball loss. While this method proved efficient, its main drawback is that it does not prevent crossing quantiles – meaning that it is possible in theory to obtain $q_1 > q_9$ (where $q_1$ is the first decile and $q_9$ is the ninth decile). To fix this, different approaches have been tried to approximate the quantiles of the distribution (Yang et al., 2019; Zhou et al., 2020), through the use of distribution distances rather than quantile loss. The work listed above takes a non-parametric approach, from a classic statistical viewpoint, as they do not assume any particular shape for the distribution. While non-parametric methods are known for their flexibility, they sometimes exhibit a high variance, depending on their smoothing parameters. Moreover, the use of non-parametric estimations of quantiles prevents aggregation of agents and their results, as one cannot simply sum quantiles. More recently, research has been conducted on robust Bayesian reinforcement learning (Derman et al., 2020) to adapt to environment changes. In this paper, the authors develop a model geared towards handling distributional shifts, but not towards handling the overall distributional outcomes of the $Q$-value.

In our paper, we consider a very flexible distribution that is parameterized by its quantiles, and from which we can both sample and extract summary characteristics (Chalabi et al., 2012).

## 3 RetaiL, An Inventory Replenishment Simulator

In this section, we detail the inner workings of the simulation environment we introduce.[4]

### 3.1 Inventory replenishment

We can frame part of the process of inventory replenishment as a *manager* passing item orders to a *warehouse* to restock a *store*. At every step, items in the store are consumed by customers. Let us consider a single item $i$ and its observation $o(i)$ with a shelf life $s_i$. We study restocking and consumption of this item over a total of $T$ time periods, each composed of $\tau \in \mathbb{N}$ sub-periods that we call time steps. During each time period $t \in \{1, \ldots, T\}$, the manager can perform $\tau$ orders of up to $n$ instances of the item $i$. Each of those orders is then added $L$ time steps later to the inventory – termed the *lead-time*. In the meantime, $\tau$ consumptions of up to $n$ items are realized by customers. Each of those purchases then results in a profit. Assuming that

---

[4]A preliminary version of the RetaiL environment was presented in (Jullien et al., 2020); the preliminary version lacked the time component in the forecast, dependency management, and detailed examples.

not enough instances of $i$ are present in the stock to meet customer demand, it then results in a missed opportunity for the manager, resulting in a loss. At the end of the period $t$, all instances of $i$ currently present in the store have their shelf life decreased by one, down to a minimum of zero. Once an instance of $i$ reaches a shelf life of zero, it is then discarded from the inventory, and creates a loss of $i$'s costs for the manager. Furthermore, the restocking and consumption of $i$ are made in a LIFO way, as customers tend to prefer items that expire furthest from their purchase date (Li et al., 2017; Cohen & Pekelman, 1978).[5]

To fulfill its task, the manager has access to a *forecast* of the customer demand for $i$ in the next $w$ time steps, contained in $o(i)$. While we could argue that the agent should be able to act without forecast, this does not hold in real-world applications. In most retail organizations, forecasts are owned by a team and used downstream by multiple teams, including the planning ones that take decisions from it. This means that the forecast is "free-to-use" information for our agent. Moreover, this means that adapting to the forecast will prove more reliable in the case of macroeconomic tail-events (lockdowns, pandemics, canal blockades, etc.) as those can be taken into account by the forecast. Obviously, this forecast is only an estimation of the actual realization of $i$'s consumption, and is less accurate the further it is from the current time step $t$.

Items are considered independent, meaning that we do not take exchangeability into account. Using this information about all individual items in the store, our goal is to learn an ordering policy to the warehouse that generalizes to all items. An ordering policy simply refers to how many units we need to order at every time step, given the context information we have about the state. The goal of our policy is to maximize overall profit, instead of simply sales. This means that waste, and missed sales are also taken into account. Moreover, while our policies have access to information about the consumption forecast of the items, this forecast is not deterministic. Indeed, some of the mechanics of the environment are hidden to the agent: the number of customers per time-step, despite being correlated to the previous time-step, is hidden, as the agent is delayed in its observation. This means that reinforcement learning agents evolve in a partially observable Markov decision process (POMDP), where an observation and an action correspond to a reward and state distribution, and not a scalar.

Formally, we can write the problem as finding a policy $\pi^* : O \to \mathbb{N}$ such that:

$$\pi^* = \arg\max_{\pi} \sum_{i \in I} \sum_{t \in T} \sum_{\tau \in t} R_\pi(o_\tau(i)), \tag{1}$$

where $o_\tau(i)$ is the observation of item $i$ at the time-step $\tau$ and $R_\pi$ the reward function parameterized by the policy $\pi$. In the following sections, we detail how we model the items, the consumption process as well as the Markov decision process we study.

### 3.1.1 Item representation

Using real-world data of items being currently sold is impossible, as it would contain confidential information (e.g., the cost obtained from the supplier). This is why we fit a copula on the data we sourced from the retailer to be able to generate what we call pseudo-items: tuples that follow the same distribution as our actual item set. Having pseudo-items also allows us to generate new, unseen item sets for any experiment. This proves useful for many reinforcement learning endeavors (Tobin et al., 2017).

When an instance of our experimental environment is created, it generates an associated set of pseudo-items with their characteristics: cost, price, popularity and shelf life. These characteristics are enough to describe an item in our setting: we do not recommend products, we want to compute waste and profit. We provide the item generation model and its parameters along with our experiments.

### 3.1.2 Consumption modelling

We model the consumption as the realization of a so-called $n, p$ process, as this way of separating the number of customers and purchasing probability is common in retail forecasting (Juster, 1966). We consider that a

---

[5]We make the assumption that the price does not depend on the remaining shelf life of the item.

day is composed of several time steps, each representing the arrival of a given number of customers in the store.

### 3.1.3   POMDP formalization

Customer consumption depends on aleatoric uncertainty, and forecast inaccuracy derives mostly from epistemic uncertainty: a forecast capable of knowing customer intent would leave little room for aleatoric uncertainty. Yet, there is no difference for our agent, as both of those uncertainties affect the reward and transitions in the environment. This means that the environment is partially observable to our agent, as the agent is unable to see past realizations of demand over linked time periods: a customer that comes in the morning will not come in the afternoon in most cases. A fully observable state would contain past realizations of the demand over the previous time steps. Formally, we can write a partially observable Markov decision process as a tuple $\langle O, A, R, P \rangle$, where $O$ is the observation we have of our environment, $A$ the action space, $R$ the reward we receive for taking that action, and $P$ the transition probability matrix. Here, they correspond to:

$O$ The full inventory position of the given item (all its instances and their remaining shelf lives), its shelf life at order, its consumption forecast, its cost and its price;

$A$ How many instances of the item we need to order; and

$R$ The profit, to which we subtract profit of missed sales and cost of waste.

### 3.2   Environment modeling

We aim to model a realistic grocery store that evolves on a daily basis through customer purchases and inventory replenishment. To do so, we rely on expert knowledge from a major grocery retailer in Europe. Our environment relies on four core components: item generation, demand generation, forecast generation, and stock update for reward computation.

**Item generation.** We define an item $i$ as a tuple containing characteristics common to all items in an item set: shelf life, popularity, retail price, and cost: $i = \langle s, b, v, c \rangle$.[6] As data sourced from the retailer contains sensitive information, we want to be able to generate items *on-the-fly*. As purchases in retail are highly repetitive, we will base ourselves on the *popularity b* of the items to generate the demand forecast in Section 3.2. On top of helping with anonymity, being able to learn in a different but similar environment has proved to help with the generalization of policies (Tobin et al., 2017). To do so, we fit a Clayton copula (Yan, 2007) on the marginal laws (gamma and log-normal) of our tuple. The parameterized model is available with the code. Given the parameterized copula, we can generate an unlimited number of tuples that follow the same multivariate distribution as the items available in the data sourced from the retailer.

**Demand generation.** To represent a variety of demand scenarios, we based the demand on the popularity of the items given by the past purchases in the real data. We then modeled a double seasonality for items: weekly and yearly.[7] Overall, given a customer visiting the store, we can write the purchase probability at a time period $t$, $p_i(t)$ for a pseudo-item $i$ as:

$$p_i(t) = b_i \cdot \cos(\omega_w t + \phi_{1,i}) \cdot \cos(\omega_y t + \phi_{2,i}), \tag{2}$$

where $b_i$ is the popularity (or base demand) for item $i$, $\phi_{.,i}$ its phases, and $\omega_w, \omega_y$ are the weekly and yearly pulsations of the demand signal, respectively. The base demand $b_i$ comes from the fitted copula, and the phases $\phi_{.,i}$ are randomly sampled to represent a variety of items. Together with the purchase probability, we also determine the number of customers who will visit the store on a given time-step. To do so, we model a multivariate Gaussian over the day sub-periods, with negatively correlated marginal laws (if a customer

---

[6]Our repository also includes dimensions, to allow for transportation cost computation.
[7]For example, beers are often sold at the end of the week, and ice cream in the summer.

comes in the morning, they will not come in the evening). Having the purchase probability and the number of customers $n(t)$, we can then simply sample from a binomial law $\mathcal{B}(n(t), p_i(t))$ to obtain the number of units $u_i$ of item $i$ sold at the time-step $t$.

**Forecast generation.**   The parameters $(n(t), p_i(t))$ of the aforementioned binomial law (Section 3.2) are not known to the manager that orders items. Instead, the manager has access to a forecast – an estimator of the parameters. We simply use a mean estimator for $n(t)$, as seasonality is mostly taken into account via our construction of $p_i(t)$.

As for the purchase probability estimator, we assume that the manager has access to a week-ahead forecast. We write the estimator as such:

$$\hat{p}_i(t + \delta_t) = p_i(t + \delta_t) + \delta_t \epsilon_i, \tag{3}$$

where $\delta_t \in \{1, \ldots, 7\}$ and $\epsilon_i \sim \mathcal{N}(0, \sigma)$. The noise $\epsilon_i$ represents the forecast inaccuracy for the item $i$, and the uncertainty about the customer behavior the manager and the store will face in the future. We assume a single $\sigma$ for all items and a mean of 0, as most single point forecasts are trained to have a symmetric, equally-weighted error. $\delta_t$ is used to show the growing uncertainty we have the further we look in the future.

**Stock update.**   To step in the environment, the agent needs to make an order of $n$ units $u_i$ of the item $i$. We consider that a time period $t$ is a succession of several time-steps.[8] At the beginning of a time period, items that were ordered $L$ time-steps before are added to the stock, where $L$ is the lead time. If the total numbers of items would exceed a maximum stock size $M$, the order is capped at $M - n$. The generated demand is then matched to the stock. Items are removed from the stock in a LIFO manner, as is the case in most of the literature (Li et al., 2017; Cohen & Pekelman, 1978). Items that are removed see their profit added to the reward. If the demand is higher than the current stock, the lacking items see their profits removed from the reward (missed sales). Finally, if the step is at the end of the day, all items in store receive a penalty of one day on their remaining shelf lives. Items that reach a shelf life of 0 are then removed from the inventory, and their cost is then removed from the reward: these items are the waste.

## 4   Inventory Replenishment Methods for Perishable Items

In the previous section, we introduced the environment we built, along with its dynamics. In this section, we introduce the baselines we consider, together with our own algorithm, GTDQN.

### 4.1   Baselines

In this subsection, we introduce the various algorithms that serve as baselines. We draw one example from the supply chain literature, as well as several from the field of Reinforcement Learning. We focus on DQN (Mnih et al., 2015) and its derivatives, as they are simple to apprehend.

$(s, Q)$ **Ordering policy.**   The $(s, Q)$ ordering policy (Nahmias & Demmy, 1981) consists of ordering $Q$ units of stock when the inventory position goes below a certain threshold $s$. While very simple, it has been in use (along with some of its derived cousins (Kelle & Milne, 1999; Cachon, 1999)) for decades in supply chain settings.

**Deep-Q-networks (DQN).**   The first reinforcement learning baseline we use is Deep-Q-Networks (Mnih et al., 2015). While this model is not SOTA anymore, it is often a reliable approach to a sequential decision-making problem, mainly in games like Atari, for instance. The idea behind DQN is to predict the Q-value of all possible actions that can be taken by the agent for a specific input. By using those values, we are able to use the corresponding policy to evolve in the environment.

---

[8]Usually, a time period would be a day, meaning that a store can be replenished several times during the course of a day.

**C51.** Categorical DQN (Bellemare et al., 2017) can be seen as a multinomial DQN with 51 categories and is a distributional version of DQN. Instead of predicting the Q value, the model divides the possible sum of future rewards interval in 51 (can be more or less) intervals. Then, the network assigns a probability to each interval, and is trained like a multinomial classifier.

**Quantile regression DQN (QR-DQN).** DQN using quantile regression (Dabney et al., 2018) is not necessarily more performant than C51. Instead of a multinomial classifier, this algorithm performs a regression on the quantiles of the distribution function. This approach has the benefit of being non-parametric, but does not guarantee that the quantiles will not cross each other: we can obtain Q10 < Q90, which would be impossible in theory. While some authors sort the obtained quantiles to remove the contradiction, we think this results in a bias in the statistics that are learnt that way.

**Expectile regression DQN.** Expectile regression DQN (ER-DQN) (Rowland et al., 2019) takes the idea behind QR-DQN and replaces quantiles with expectiles. It is possible to interpret an expectile as the "value that would be the mean if values above it were more likely to occur than they actually are" (Philipps, 2022).

### 4.1.1 Underlying neural architecture

All the considered DQN-based algorithms, including the following GTDQN, are based on the same feed-forward architecture. The individual shelf lives of the already stocked items are first processed together in a convolution layer. They are then concatenated with the item characteristics and processed through a simple Feed-Forward Deep Neural Network with Layer Norm and SELU activation (Klambauer et al., 2017).

### 4.2 Generalized Tukey deep Q-network

In this section, we introduce a new baseline, *generalized Tukey deep Q-network* (GTDQN), for decision-making in stochastic environments. As the problem we study requires planning under uncertainty, we need a baseline that can consider randomness in the signals it receives from the environment. While we can use classic off-policy architectures like Deep Q Networks, the partial observability of our environment is more likely to be encompassed by an algorithm that assumes value distributions over actions rather than simple scalar values.

Thus, we assume that the $Q$-value follows a generalized lambda distribution, also known as a generalized Tukey distribution. This is a weak assumption that does not constrain the shape too much. Indeed, the use of four parameters allows for a very high degree of flexibility of shapes for this distribution family (Chalabi et al., 2012): unimodal, s-shaped, monotone, and even u-shaped. The generalized lambda distribution can be expressed with its quantile function $\alpha$ as follows:

$$\alpha_\Lambda(u) = \lambda_1 + \frac{1}{\lambda_2}[u^{\lambda_3} - (1-u)^{\lambda_4}], \tag{4}$$

where $\Lambda = (\lambda_1, \lambda_2, \lambda_3, \lambda_4)$ is the tuple of four parameters that define our distribution. Those parameters can then be used to compute the distribution's four first moments (mean, variance, kurtosis and skewness), if they are defined.

Thus, we build our $Q$-network not to predict the expected $Q$-value nor its quantiles, but to predict the parameters $\lambda_1, \lambda_2, \lambda_3, \lambda_4$ of a generalized lambda distribution. This allows us to obtain both guarantees on the behavior of the distribution's tail, and non-crossing quantiles. Still, we perform our Bellman updates by estimating the quantiles derived from the values of the distribution's parameters. To obtain a quantile's value, we simply need to query it by using Equation 4.

Working with quantiles guarantees that the Bellman operator we use is a contraction, when using a smoothed pinball loss (Yang et al., 2019). While written differently in most of the literature, the classic pinball loss can be written as follows:

$$\mathcal{PL}_u(y, \hat{y}(u)) = (y - \hat{y}(u)) \cdot u + \max(0, \hat{y}(u) - y), \tag{5}$$

---

**Algorithm 1:** Generalized Lambda Distribution Q-Learning

**Require:** quantiles $\{q_1, \ldots, q_N\}$, parameter $\delta$

**Input** : $o, a, r, o', \gamma \in [0, 1]$

1 $\Lambda(o', a'), \forall a' \in A$         # Compute distribution parameters ;

2 $\Lambda^* \leftarrow \arg\max_{a'} \hat{\mu}(\Lambda(o', a'))$   # Compute optimal action (Equation 7) ;

3 $\mathcal{T}q_i \leftarrow r + \gamma\alpha_{\Lambda^*}(q_i), \forall i$       # Update projection via Equation 4 ;

4 Optimize via loss function (Equation 6) ;

**Output :** $\Sigma_{j=1}^N \mathbf{E}_i[\mathcal{L}_{q_j}^\delta(\mathcal{T}q_i, \alpha_{\Lambda(o,a)}(q_j))]$

---

where $u$ is a quantile, $y$ the realized value, and $\hat{y}(u)$ the predicted value of quantile $u$. Its $\delta$-smoothed version is obtained by plugging this loss estimator instead of the square error in a Huber loss (Huber, 1992). This gives us the following loss function:

$$\mathcal{L}_u^\delta(y, \hat{y}_\Lambda(u)) = \begin{cases} \frac{1}{2}\left[y - \hat{y}_\Lambda(u)\right]^2 \Delta), & \text{for } |y - \hat{y}_\Lambda(u)| \leq \delta \\ \delta\left(|y - \hat{y}_\Lambda(u)| - \delta/2\right)\Delta), & \text{otherwise}, \end{cases} \tag{6}$$

with $\Delta = \mathcal{PL}_u(y, \hat{y}_\Lambda(u))$, where $\delta$ is a smoothing parameter and $u$ the considered quantile for the loss. Algorithm 1 shows the way we update the parameters of our network through temporal difference learning adapted to a quantile setting.

Unlike C51 (Bellemare et al., 2017) and QR-DQN (Dabney et al., 2018), we do not select the optimal action (line 2 of Algorithm 1) via an average of the quantile statistics, but via a mean estimator obtained via our GLD distribution's parameters (Fournier et al., 2007):

$$\hat{\mu}(\Lambda) = \lambda_1 + \frac{\frac{1}{1+\lambda_3} - \frac{1}{1+\lambda_4}}{\lambda_2}. \tag{7}$$

This approach is closer to the implementation of ER-DQN (Rowland et al., 2019), where only the expectile 0.5 is used, rather than QR-DQN, where the quantiles are averaged to obtain an estimation of the mean (Dabney et al., 2018).

## 5 Experiments

In this section, we compare the performance in inventory replenishment simulation of our new baseline against a number of baselines (RQ3), for a variety of scenarios. We want to see whether we can improve overall profit (RQ1), and, if so, if it comes at the cost of generating more waste (RQ2).

### 5.1 Experimental setup

We train our DQN-family policies (baselines and GTDQN) on a total of $6\,000$ pseudo-items, for transitions of $5\,000$ steps. We do so in order to expose our agents to a variety of possible scenarios and items. Morover, we do not train our agents in an average reward framework, as discounting also presents an interest for accounting in supply chain planning (Beamon & Fernandes, 2004). We evaluate the performance of our agents on a total of 30 generations of 100 unseen pseudo-items, for $2\,000$ steps. We repeat this for 3 different scenarios of randomness, indicating how observable the environment is. We name them $H = 0$, $H = 1$, $H = 2$:

$H = 0$ In this scenario, the environment's mechanics are not random. Here, Equation 3 reduces to $\hat{p}_i(t+\delta_t) = p_i(t + \delta_t)$. This means that the agent knows exactly the purchase probability of items. In this scenario, there is little need for adaptability as the inter-day variations in customer behavior are close to non-existent.

$H = 1$ In this scenario the environment's mechanics are slightly random and overall exhibit little variation. In this scenario, the agent needs to learn how to interpret the week-ahead forecast and leverage it to increase profit.

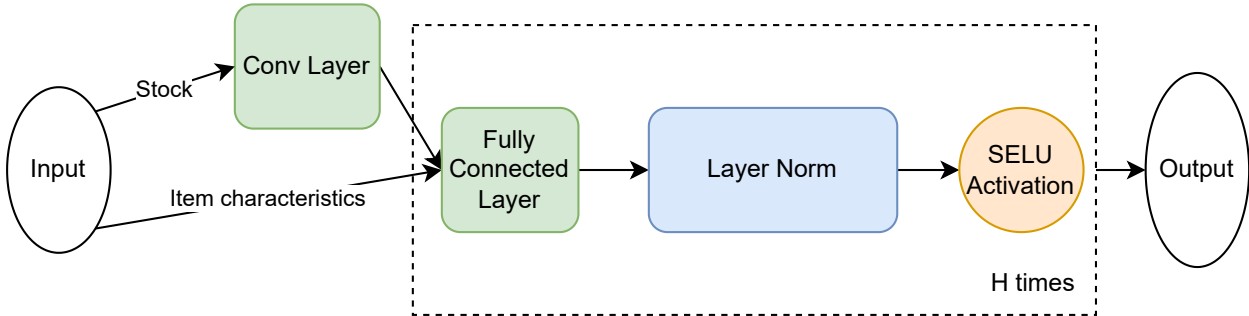

Figure 1: Neural architecture for all considered DQN-based models

$H = 2$ In this scenario, the environment is highly noisy and becomes much harder to predict.

These scenarios allow us to verify whether an agent has learned a decent policy and is able to generalize to unseen data. Real grocery stores with a "good" forecast are more likely to be represented by the $H = 1$ and $H = 2$ scenarios (Ramanathan, 2012). The agents are trained on a reward function $R$ defined as $R = Sales - Waste$. In Section 5.2.3, the reward function used is $R = Sales - 10 \times Waste$ in order to assess how well agents reduce waste when given a new target.

We perform two experiments, where we look at overall profit performance and waste reduction relative to a baseline, respectively.

**Experiment 1: Impact of forecast inaccuracy.** In this experiment, we measure the overall performance of the various agents, for the different levels of environment randomness (RQ1). This experiment allows us to measure the impact of randomness and unpredictability of consumption behavior on our agents, and to see whether they are an improvement over a deterministic heuristic.

**Experiment 2: Impact of unstable order behavior on waste.** In this experiment, we show how the orders translate into generated waste. This way, we can see whether the improvement in the previous section comes at the cost of more waste or not (RQ2).

**Implementation and computational details.** Our code was implemented in PyTorch (Paszke et al., 2019) and is available on GitHub.[9] We ran our experiments on a RTX A6000 GPU, 16 CPU cores and 128GB RAM. All models use the same underlying neural network architecture as shown in Figure 1. We notice that GTDQN was approximately 3 times faster than QR-DQN and ER-DQN for more than 4 quantiles, as it needs estimating a constant number of parameters. We performed a grid search on DQN for all hyperparameters, and kept those for all models. However, we set the exploration rate at 0.01 for distributional methods, following (Dabney et al., 2018). Training curves are available in Appendix A.2.

## 5.2 Results

In this section, we detail the performance of the various baselines as well as GTDQN, introduced in Section 4.2, for both resistance to uncertainty and waste reduction. We averaged the results of the different algorithms over a total of 6,000 pseudo-items.

### 5.2.1 Overall performance

We report the performance in Table 1 as the improvement relative to a simple $(s, Q)$ policy, as this kind of policy is still prominent in supply chain practices (Jalali & Van Nieuwenhuyse, 2015). In this table, we see that all models perform better than the baseline when there is no uncertainty ($H = 0$). Yet, there is no significant difference between them.

---
[9] https://github.com/samijullien/GTDQN

Table 1: Human-normalized profit (higher is better). Results on trajectories of length $2\,000$, averaged over $3\,000$ items, for 3 different consumption volatility scenarios. Bold indicates best.

| | Quantiles | $H = 0$ | $H = 1$ | $H = 2$ |
|---|---|---|---|---|
| DQN | – | 146.1% ±0.7 | 178.8% ±0.5 | 146.5% ±0.3 |
| C51 | – | 142.7% ±0.9 | 173.0% ±0.4 | 156.1% ±0.2 |
| QR-DQN | 5 | 146.7% ±0.7 | 190.4% ±0.9 | 176.6% ±0.6 |
| | 9 | 147.2% ±0.8 | 204.2% ±0.11 | 189.5% ±0.6 |
| | 15 | 146.7% ±0.1 | 193.3% ±0.8 | 161.6% ±0.4 |
| | 19 | 146.1% ±0.1 | 198.7% ±0.8 | 170.1% ±0.5 |
| ER-DQN | 5 | 147.4% ±0.9 | 203.5% ±0.9 | 172.5% ±0.5 |
| | 9 | 145.1% ±0.7 | 177.7% ±0.6 | 174.2% ±0.5 |
| | 15 | **148.7%** ±0.9 | 202.2% ±0.8 | 168.0% ±0.5 |
| | 19 | 146.1% ±0.1 | 209.3% ±0.9 | 170.3% ±0.6 |
| GTDQN (ours) | 5 | 147.8% ±0.8 | **213.3%** ±0.9 | 186.2% ±0.5 |
| | 9 | 147.9% ±1.1 | 208.3% ±1.0 | **194.1%** ±0.7 |
| | 15 | 143.1% ±0.6 | 209.5% ±0.9 | 192.8% ±0.6 |
| | 19 | 147.3% ±0.9 | 212.6% ±0.9 | 191.8% ±0.6 |

In the second scenario with medium volatility ($H = 1$), the performance improvement of distributional methods over deterministic ones shows clearly, highlighting the performance of QR-DQN, GTDQN and ER-DQN. C51 exhibits a performance closer to DQN than to the other distributional approaches. It is additionally much slower to train than all others. C51 being unable to update its bucket values might be a reason why its performance is slightly disappointing – yet, it still is a clear improvement over the $(s, Q)$ baseline.

In the third scenario ($H = 2$), it is made even more obvious that the non-bounded distributional approaches can capture the uncertainty, as they widen the gap with the more simple DQN. Our method, GTDQN, is overall better than ER-DQN, that bases itself on expectiles. Our method is relatively more stable with respect to how many quantiles or expectiles it estimates with. Moreover, its computation time is much lower than ER-DQN and QR-DQN for $N > 4$, as it does not estimate new parameters.

Looking closer at the results in Figure 2, we can see that improvements in profit by GTDQN relative to the baseline are strictly one-sided in high-entropy scenarios. Moreover, they show that the resulting distribution of improvements is not a gaussian – this is why the MAD was preferred as a metric for uncertainty quantification. This means that using GTDQN results in a consistent improvement in profit performance.

### 5.2.2 Waste reduction

In Table 2, we visualize the waste generated relative to our simple $(s, Q)$ policy baseline. In all scenarios, we see that all methods reduce waste relative to the baseline. This means that they managed to improve the overall score (Table 1), without increasing waste: they ordered more than the baseline, and wasted less products. This is not surprising: the baseline only considers the number of items in the stock, not when they expire. Learning this both contributes to increased score and reduced waste. In all scenarios, we see that all methods reduce waste relative to the baseline.

In the scenario with full information, C51 performs very well, followed closely by GTDQN and some verisons of ER-DQN. In the the $H = 1$ scenario, where the environment is partially observable, both GTDQN and QRDQN perform similarly. Surprisingly, C51 does not perform well and is the worst of all models considered here. Given the significant improvement over the baseline in a partially observable environment brought by those methods, we conclude that they were able to adapt to the environment's dynamics and its randomness, while still taking the potential waste into account. Finally, in the $H = 2$ scenario, all models perform comparably well.

Table 2: Human-normalized waste (lower is better). Results on trajectories of length 2 000, averaged over 3 000 items, for 3 different consumption volatility scenarios. Bold indicates best.

|  | Quantiles | $H = 0$ | $H = 1$ | $H = 2$ |
|---|---|---|---|---|
| DQN | – | 16.8% | 23.0% | 13.2% |
| C51 | – | **2.6**% | 66.9% | 17.3% |
| QR-DQN | 5 | 32.1% | 16.5% | **9.6%** |
|  | 9 | 46.7% | 13.9% | 12.6% |
|  | 15 | 20.1% | 17.7% | 16.3% |
|  | 19 | 13.8% | 19.4% | 11.1% |
| ER-DQN | 5 | 46.5% | 26.7% | 13.3% |
|  | 9 | 6.1% | 23.4% | 12.8% |
|  | 15 | 8.4% | 14.5% | 11.5% |
|  | 19 | 30.1% | **13.1%** | 11.3% |
| GTDQN (ours) | 5 | 15.6% | 13.8% | 14.6% |
|  | 9 | 8.1% | 19.3% | 16.2% |
|  | 15 | 4.9% | 14.5% | 16.3% |
|  | 19 | 6.1% | 13.9% | 15.9% |

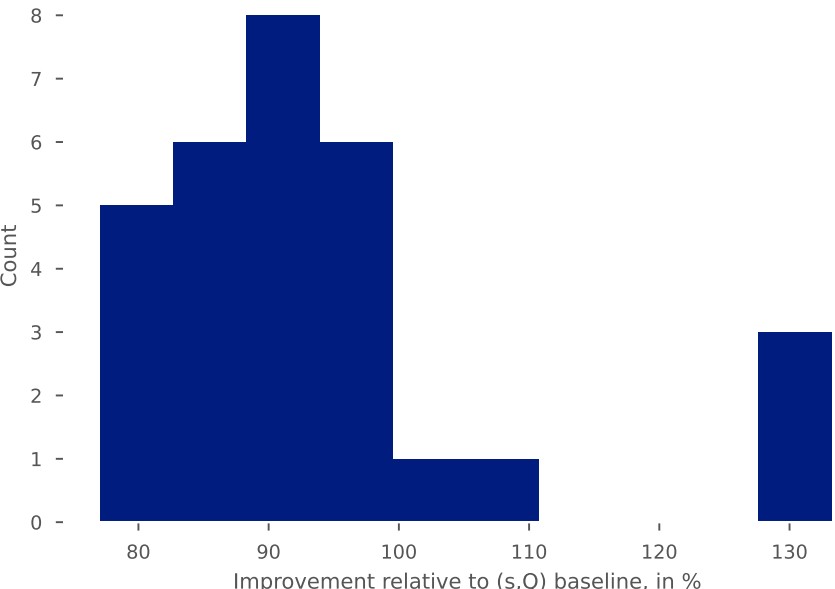

Figure 2: Improvement of GTDQN over $(s, Q)$-policy, for the $H = 2$ scenario, for 30 generations of 100 items.

Note that GTDQN is constantly in the same neighborhood as the best solution, no matter the number of computed quantiles or the randomness of the environment.

### 5.2.3 Waste reduction for waste-averse agents.

In this section, we look at the performance of all evaluated algorithms, trained under a more waste-averse reward. We evaluated the algorithms with the same setup, and a Reward function $R = Sales - 10 \times Waste$.

Table 3: Comparison of normalized performance in profit and generated waste, for a higher weight on waste during learning. (bold indicates best). Results on transitions of length $2\,000$, averaged over $6\,000$ items, for 3 different consumption volatility scenarios.

|  | Profit | | | Waste | | |
|---|---|---|---|---|---|---|
|  | $H = 0$ | $H = 1$ | $H = 2$ | $H = 0$ | $H = 1$ | $H = 2$ |
| DQN | **140.9**$\% \pm 0.8$ | $179.4\% \pm 0.4$ | $158.3\% \pm 0.3$ | $8.6\%$ | $28.9\%$ | $13.9\%$ |
| C51 | $139.3\% \pm 0.8$ | $175.1\% \pm 0.7$ | $152.3\% \pm 0.3$ | $24.5\%$ | $89\%$ | $27.7\%$ |
| QR-DQN@9 | $138.6\% \pm 1.0$ | $185.1\% \pm 0.7$ | $162.5\% \pm 0.3$ | $26.1\%$ | $14.3\%$ | $13.6\%$ |
| ER-DQN@9 | $138.5\% \pm 1.1$ | $167.3\% \pm 0.5$ | $163.9\% \pm 0.3$ | $23.9\%$ | **11.6**$\%$ | $8.9\%$ |
| GTDQN@9 (ours) | $140.3\% \pm 1.1$ | **186.3**$\% \pm 0.7$ | **164.4**$\% \pm 0.3$ | **4.1**$\%$ | $17.3\%$ | **8.8**$\%$ |

Table 4: Comparison of normalized performance in profit and generated waste of GTDQN with and without Equation 7 (bold indicates best). Results on transitions of length $2\,000$, averaged over $6\,000$ items, for 3 different consumption volatility scenarios.

|  | Profit | | | Waste | | |
|---|---|---|---|---|---|---|
|  | $H = 0$ | $H = 1$ | $H = 2$ | $H = 0$ | $H = 1$ | $H = 2$ |
| GTDQN without Equation 7 | $141.5\%$ | $150.5\%$ | $132.1\%$ | $93.6\%$ | $39\%$ | $40.1\%$ |
| GTDQN | **147.8%** | **213.3%** | **186.2%** | **15.6%** | **13.8%** | **14.6%** |

Table 2 shows that all agents do reduce their sales, and that it mostly comes at a lower waste, compared to Table 2. We can see from the table that GTDQN maintains its lead in profit, and also reduces its generated waste on every scenario compared to the previous reward signal.

### 5.2.4 Impact of the mean estimator

It is of interest to know why GTDQN performs well on the *RetaiL* environment, despite it not being tuned for it. We thus perform an ablation study, where we estimated our parameter vector $\Lambda = \langle \lambda_1, \lambda_2, \lambda_3, \lambda_4 \rangle$. However, instead of using Equation 7 to select the optimal action, we compute 5 quantiles and average them to estimate the mean, as it is the case in QR-DQN. As shown in Table 4, our mean estimator in Equation 7 has a strong impact, both on waste and profit.

In conclusion, we have shown that it is possible to improve the restocking strategy for perishable items by using a distributional algorithm (RQ1). Moreover, this improvement in overall profit does translate to lower waste (RQ2). Finally, we have shown that the algorithm we propose, GTDQN, does present a strong alternative to other distributional algorithms, as it is constant in its good performance (RQ3).

## 6 Conclusion

In this paper, we have introduced a new reinforcement learning environment, *RetaiL*, for both supply chain and reinforcement learning practitioners and researchers. This environment is based on expert knowledge and uses real-world data to generate realistic scenarios. By taking waste at the item-level into account, and by being able to tune the forecast accuracy as well as the customer's behavior, we can act on the environment's noisiness; this results in a partially observable MDP, with tunable stochasticity, which is lacking for most RL tasks. Inventory management in *RetaiL* needs the agent to pick up seasonal patterns, unpredictability of customer demand, as well as delayed action effects, and credit assignment as it works in a LIFO manner.

Additionally, we have proposed Generalized Tukey Deep Q Networks (GTDQN), a new algorithm aimed at estimating a wide range of distributions, based on DQN. GTDQN offers the consistency of parameterized distributions, but can be trained by quantile loss instead of likelihood-based approaches. Moreover, GTDQN can represent a wide array of distributions, and does not suffer from the quantile crossing phenomenon. We

have found that GTDQN outperforms other methods from the same family in most cases for the task replenishment of perishable items under uncertainty. GTDQN does so by using a quantile loss to optimize a well-defined distribution's parameters and selecting optimal actions using a mean estimator. GTDQN does not require any assumptions specific to the simulation environment we provide. We have also found that GTDQN can offer significant and constant improvement over our classic supply chain baseline, as well as over other distributional approaches, outperforming ER-DQN in highly unpredictable environments. Moreover, GTDQN does this without generating more waste through its replenishment policies, hinting that it learnt the environment's dynamics better than the baselines. Our results point towards distributional reinforcement learning as a way to solve POMDPs.

As to the broader impact of our work, the simulation environment we provide with the paper rewards weighting the risks of wasting an instance of an item and the profit from selling it. This might favor resupply of stores in more wealthy geographical areas where the average profit per item is higher. Thus, any deployment of such an automated policy should be evaluated on different sub-clusters of items, to ensure it does not discriminate on the purchasing power of customers. This kind of simulation can be replicated for all domains that face uncertainty and inventory that lowers in perceived quality with time – for instance, the fashion industry.

A limitation of our work is that we only considered discrete action spaces, whereas our environment would more be adapted to infinite-countable ones. Moreover, we consider marginal demand between items to be independent, which is unlikely to be the case in real life. Finally, our environment, *RetaiL*, assumes no cost to restock. This most likely inflates slightly the performance of the algorithms we consider for stores that do not have scheduled restocking as the ones we consider. Furthermore, a proper implementation in production would require a continuous control mechanism (such as Model Predictive Control) to match the desired stock level.

For future work, we intend to model price elasticity of customers in order to model item consumption in case of out-of-stock items. We also want to add a restocking cost based on volume and weight of items. We plan to extend GTDQN for multi-agent reinforcement learning, as our estimation of parameters gives us access to cumulants that can be used to sum rewards of various agents and policies. This would be especially interesting given the low number of parameters in GTDQN. Furthermore, we plan to extend it to continuous action spaces, to leverage the structure of the data more efficiently. Finally, we plan to study whether those automated replenishment policies based on balancing profits and waste do not disadvantage some categories of customers more than others.

## Acknowledgements

We want to thank our anonymous reviewers for helping us improve this paper. This research was supported by Ahold Delhaize and the Hybrid Intelligence Center, a 10-year program funded by the Dutch Ministry of Education, Culture and Science through the Netherlands Organisation for Scientific Research. `https://hybrid-intelligence-centre.nl`.

All content represents the opinion of the authors, which is not necessarily shared or endorsed by their respective employers and/or sponsors.

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

# A  Appendix

## A.1  Environment logic

We present a few code examples of the environment we provide in Section 3. Listing 1 shows the global stepping logic of the environment. Listing 2 shows the addition of items in the environment's stock: as individual items are represented by their shelf life, items are added via the sorting of the current stock to prevent item "mix-up". Finally, Listing 3 shows part of the reward computation, when items are sold and removed from the stock matrix.

Listing 1: Stepping logic of the environment

```python
def step(self, action):
    #Put bounds on action
    new_action = (
        torch.as_tensor(action, dtype=torch.int32)
        .clamp(0, self._max_stock)
        .to(self.device)
    )
    #If this is a new day, we take waste into account
    if self.day_position % self._substep_count == 0:
        order_cost = self._make_fast_order(new_action)
        (sales, availability) = self._generateDemand(self.real.clamp_(0.0, 1.0))
        waste = self._waste()  # Update waste and store result
        self._reduceShelfLives()
        self._step_counter += 1
        self._updateEnv()
    else:
        self.day_position += 1
        order_cost = self._make_order(new_action)
        (sales, availability) = self._generateDemand(self.real.clamp_(0.0, 1.0))
        waste = torch.zeros(
            self._assortment_size
        )  # By default, no waste before the end of day
        self._updateObs()
    sales.sub_(order_cost)
    self.sales = sales
    self.total_sales += sales
    self.waste = waste
    self.total_waste += waste
    self.availability = availability
```

Listing 2: Addition of items to the stock

```python
def _addStock(self, units):
    #Create padding
    padding = self._max_stock - units
    replenishment = torch.stack((units, padding)).t().reshape(-1)
    #Create vectors of new items
    restock_matrix = self._repeater.repeat_interleave(
        repeats=replenishment.long(), dim=0
    ).view(self._assortment_size, self._max_stock)
    # Add new items to stock
    torch.add(
        self.stock.sort(1)[0],
        restock_matrix.sort(1, descending=True)[0],
```

```
        out=self.stock,
    )
    return
```

Listing 3: Selling units and updating stock

```python
def _sellUnits(self, units):
    #Get the number of sales
    sold = torch.min(self.stock.ge(1).sum(1).double(), units)
    #Compute availability
    availability = self.stock.ge(1).sum(1).double().div(units).clamp(0, 1)
    #Items with no demand are available
    availability[torch.isnan(availability)] = 1.0
    #Compute sales
    sales = (
        sold.mul_(2)
        .sub_(units)
        .mul(self.assortment.selling_price - self.assortment.cost)
    )
    (p, n) = self.stock.shape
    #Update stock
    stock_vector = self.stock.sort(1, descending=True)[0].view(-1)
    to_keep = n - units
    interleaver = torch.stack((units, to_keep)).t().reshape(2, p).view(-1).long()
    binary_vec = torch.tensor([0.0, 1]).repeat(p).repeat_interleave(interleaver)
    self.stock = binary_vec.mul_(stock_vector).view(p, n)
    return (sales, availability)
```

## A.2 Training Curves

Figures 3 and 4 show the raw training scores of the models presented in Section 5.2.

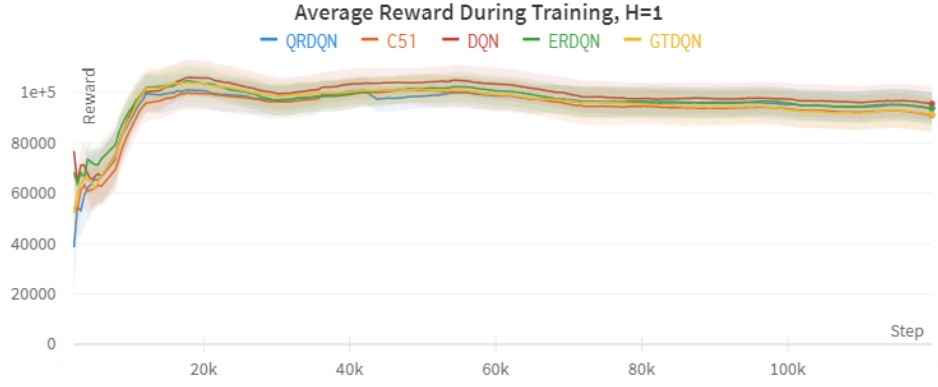

Figure 3: Raw scores during training for $H = 1$.

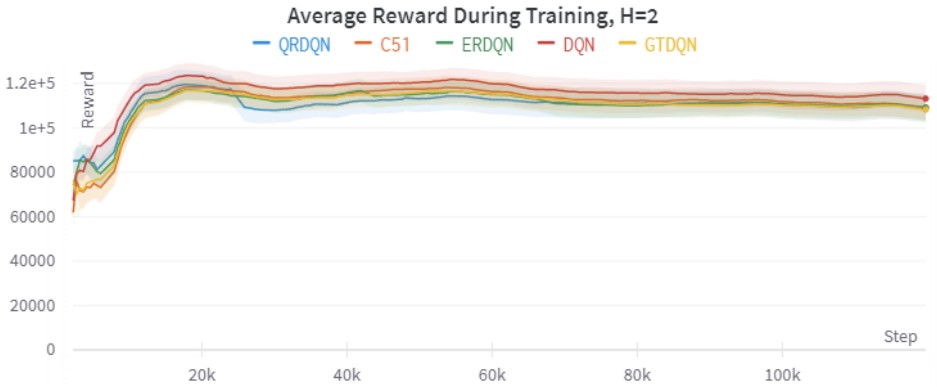

Figure 4: Raw scores during training for $H = 2$.

