# OpenReview forum: "A Simulation Environment and Reinforcement Learning Method for Waste Reduction"
_TMLR — Accepted by TMLR_

### Review · Reviewer_pVwr · 2023-02-09

**Summary Of Contributions:**

The proposed work presents a new domain for benchmarking (RL) algorithms to manage perishable inventory in a shop/warehouse. The goal in this domain is to maximize profits and minimize waste. Observations correspond to features of the product (cost price, selling price, shelf life, popularity) and also an approximate forecast of its demand. These features are derived from (anonymized) realistic data. Actions correspond to the number of new units to order. Items get replenished based on order and get consumed based on a consumption model.

Apart from introducing the domain, the paper also introduces a variant of the distributional RL algorithm, that uses the generalized lambda distribution to parameterize the return distribution. Empirical results suggest that this provides a strong baseline for the proposed domain.

**Audience:**

Yes

**Broader Impact Concerns:**

I believe that the authors have done a good job of discussing these.

**Claims And Evidence:**

Yes

**Requested Changes:**

*R1: Code documentation* The code could benefit a lot from proper documentation. While this wouldn't be a major point of concern for most papers, the proposed work is explicitly about developing an environment that others can use to test their algorithms. Currently, the submitted code lacks clear documentation.

*R2: Visualizing/interpreting the results + reward hacking* Personally, having implemented a supply-demand domain based on queueing theory, I have found that implementing a new domain is often prone to subtle bugs. These bugs get exploited by RL agents to obtain scores better than non-adaptive/fixed/classical policies taken from the literature. Unfortunately, as a reviewer, it is hard to assess the correctness of domains. Therefore, I was wondering if the authors could provide some more interpretable results/visualization of what the agent is doing when it achieves significantly better than the baseline. This could not only help identify potential bugs but could also provide insights into how to manage inventory.

*R3: Formal modeling equations* It would be very helpful to the readers if the modeling (or the core code) for the domain can be presented in the appendix of the paper.

*R4: Describe the reward function:* I suspect that the reward function is just a weighted combination of profits, cost of waste, and other variables. Depending on the combination of the weights (i.e., how much importance does reducing waste have in the reward), I suspect that the results in table 2 can change significantly. Since this relates to a primary research question RQ2 posed by the paper, is it possible to provide an ablation study for that?

**Strengths And Weaknesses:**

*Strengths*

S1: A domain that is both practically relevant and could also be used as a benchmark for testing RL algorithms

S2: A simple variant of the distributional RL method using the (fully parameterized) generalized lambda distribution that seems to work well on the presented domains.

S3: There are some missing details, but otherwise, the document is well-written.

*Weaknesses*

W1: The main weakness of the domain is related to the point of marginal demand of items being independent. Often even if one item is not there, customers might buy a related item. I suspect if each item had a feature/covariate then one could model similarity between the items as well.

W2: See the requested changes section below.

*Questions*

Q1.  Why was the episodic setting chosen instead of the average reward setting? For these problems, it seems like the average reward is more useful, because discounting is not meaningful here, and having episodes as long as 5000 timesteps can blow up the return values (unless additional care is taken to adjust the reward magnitude or the number of timesteps). Further, how is a stock update and lead time handled toward the end of an episode in this episodic setting?

Q2: How were the parameters of eqn 2 chosen?

Q3: How do the modeling equations change when H=0?

Q4: The time in purchase probability corresponds to 'time-step' or to 'time period'?

Q5: How to compute the value of a given quantile when using the generalized lambda distribution?

Q6: Experimental setup says that methods are evaluated on 100 unseen pseudo-items but Tables 1 and 2 show results on the 6000 items that were used during training.

Q7: "(H = 0). Yet, there is no significant difference between them": Can the stderr be provided in Table 1 to deduce these?

Q8: GLDQN: alation study for hyper-parameter sensitivity? How were hyper-parameters for the baseline and proposed method chosen?

Q9: I am a little confused by how the term POMDP is used in this paper. Just because the transition and reward are stochastic, one need not need to model it as a POMDP (of course, one could consider the seed of the random variable as the hidden variable. If it was observed then everything in the domain would be deterministic). Till the distribution of the next state and reward is independent of the history, conditioned on the current observation and action, then one could still model it as an MDP. If this is not the case, then I would suggest authors to avoid mentioning the contraction of the distributional Bellman operator as it need not be a contraction in the POMDP case (even if state/action sets are finite).

Minor:

- Turn on line numbers for ease of review feedback
- In conclusion: " as it works in a FIFO manner" -> " as it works in a LIFO manner"?
- Should "experiments setup" and Table 1 have "transitions of length …" replaced with "trajectories of length .."?
- "Learning the expectation of the Q-value is the most straightforward way…": Q-value *is* the expected return.
-  Is there a maximum stock capacity?
- "Yet, there is no difference for our agent, as both of them affect the reward and transitions in the environment": Why dos forecast effect the transition probability?

---

> ### Author Response · Authors · 2023-02-22
> **Response to Reviewer pVwr**
>
> Dear Reviewer pVwr,
>
> We want to thank you for taking the time to assess our work and for providing such a detailed review!
>
> First, we want to acknowledge the first weakness you mention: our environment would indeed benefit a lot from joint demand distributions. We will point this out in the limitations and future work paragraphs of our conclusion.
>
> Second, we will answer your questions about our work according to their order - we will of course detail these questions in the revision of the paper.
>
> -- Q1: You are right that discounting has limited interest in this use case. Still, we decided to use it as some supply chain planners make use of a discounting factor when computing revenue: cash now is more important than cash in the next accounting quarter (https://doi.org/10.1080/09537280410001697701); we will add this motivation and reference to the paper. As the high length of the episodes, it comes from our will to show possible wastage of every item considered. Indeed, shelf life is very high in products like pasta, for example. Moreover, this allows the agent to be exposed to several cycles of waste, which should help in credit assignment.
>
> -- Q2: The base demand $b_i$ comes from the multivariate copula we fitted on the items. The phases $\phi_{.,i}$ are randomly sampled to account for a bigger variety of seasonalities in the items. Finally, the pulsations are identical for all items, to represent weekly and yearly seasonality in item demand. We will add these clarifications to Section 3.2.
>
> -- Q3: In the case H=0, the equation (3) reduces to $ \hat{p}_i(t+dt) = p_i(t+dt)$. Following this, it means that the agent knows exactly the purchase probability of items. We will add this clarification to Section 5.1.
>
> -- Q4: In the purchase probability, the index t refers indeed to the time period. We will add this clarification in section 3.2.
>
> -- Q5: To compute the value of a given quantile q using the generalized distribution, we need the values of the four lambda parameters of the distribution we consider. We then make use of equation (4) and obtain the quantile value. We will add this clarification to Section 4.2.
>
> -- Q6: There is indeed a mistake in Tables 1 and 2. Evaluations are done over 30 sets of 100 items, hence 3000 - not 6000. Sorry about this mistake, and thanks a lot for noticing. We will of course correct tables 1 and 2 accordingly.
>
> -- Q7: We did not provide the standard errors as the distribution of values is not normal, and wanted to avoid mis-interpretations of our results.
> Here are the mean absolute deviations of Table 1 for the case H=0, in the right order: (0.07, 0.09, 0.07, 0.08, 0.10, 0.10, 0.09, 0.07,0.09, 0.09, 0.08, 0.11, 0.06, 0.09). We will add this to the paper, in Section 5.2.
>
> -- Q8: For the choice of the hyperparameters, we performed a grid search for DQN on several layer numbers and sizes, then used those on all networks considered. As for the exploration rate, we used 0.05 for DQN and 0.01 on distributional approaches, as per QR-DQN. We will add these clarifications to Section 5.1.
>
> -- Q9: While we do believe that the underlying mechanism is a POMDP (if a customer does their groceries for the week on a Thursday, they will not do them on Friday), we do indeed model it as an MDP in this environment. We will rephrase this in Section 3.1, as well as expand the related work accordingly.
>
> We also thank you for your minor comments, as they are indeed correct. To answer the two questions, there is indeed a maximum stock. An agent cannot order above this maximum stock and can only match it. We will add this clarification to Section 3.2.
>
> As for the forecast affecting the transition probability, this is indeed wrong overall, although the agent can derive a link between the two, as the forecast gives an indication in this specific case.
>
> Finally, on the requested changes:
>
> -- R1: We started documenting the code more according to your remarks. We are adding docstrings for every function, including usage examples.
>
> -- R2: We think that plotting a histogram of average order over average forecasted demand will help in alleviating the concern of the agent benefiting from a loophole in the environment. We are currently relaunching experiments to include this in Section 5.
>
> -- R3: We will add the code in the Appendix of the paper.
>
> -- R4: You are indeed correct, the reward is simply profit minus waste. Still, we allow users of the environment to define their own utility function, possibly non-linear, by defining a weighted combination of sales, waste, and availability, as the latter is usually a very important factor for store managers. We will add this clarification to Section 3, in a new subsection. We are currently rerunning the experiments with a higher weight on waste.

---

### Review · Reviewer_w2jW · 2023-02-22

**Summary Of Contributions:**

This paper presents a new simulation environment for inventory management and proposes a new type of Q-learning algorithm to solve it. The new environment code has also been released.

**Audience:**

Yes

**Broader Impact Concerns:**

None.

**Claims And Evidence:**

Yes

**Requested Changes:**

1) Can you clarify why the environment is a POMDP rather than simply an MDP? What information about the state is missing from the observation?
2) Why are you modeling the Q-values as a lambda distribution? I don't see a justification for that choice.
3) On the name 'Generalized lambda deep Q-network' - I wonder if there is any way to avoid overloading the 'lambda' here. I understand that you are using the lambda distribution, but for most RL people lambda binds very strongly to'lambda-returns', eg., Q-lambda, and I think this may cause confusion.
4) Figure 1 'all DQN-based models', presumably you mean in the paper, not all published.
5) Why are you using SELU actuvations?
6) Figure 2 is also not giving us much.
7) "Instead of predicting the Q value, the model divides the possible reward interval in 51 (can be more or less) intervals." Is it the reward or the Q-values?
8) Why are the summations in Eq. 1 so small?
9) Why does ' forecast inaccuracy derive from epistemic uncertainty'? Surely there is aleatoric noise in the process being forecasted too.
10) Please add learning curves to the results section.
11) Please add an optimal control baseline.

**Strengths And Weaknesses:**

Strengths:
1) A new RL test environment has been defined and released in code.
2) Some baseline performances have been established.

Weaknesses:
1) Some clarity issues in the writeup as I document below.
2) I'm not sure how impactful this work is, and I don't think the algorithmic contribution is major. Perhaps the environment will turn out to be useful though I am not sure.
3) How would a simple optimal-control baseline perform (eg, MPC or similar)? This is presumably how this type of problem is dealt with in practice.
4) I don't really understand the results. I would like to see some learning curves for the various methods.

---

> ### Author Response · Authors · 2023-03-01
> **Response to Reviewer w2jW**
>
> Dear Reviewer w2jW,
>
> We thank you for taking the time to go over our work.
>
> We first want to address the weaknesses you mention:
>
> W1: Thank you for pointing out the clarity issues. We hope we addressed them through the changes spelled out below.
>
> W2: We believe that modeling Q-values as a lambda distribution brings a different viewpoint on distributional approximations via the use of a fully-defined quantile function. We are learning parameterized distributions via methods coming from non-parametric approaches. As for the impact, reducing waste in supermarkets supply chains would clearly help in reaching carbon targets worldwide. While our paper does not single-handedly solve it, we believe that this is a step in the right direction.
>
> W3: Optimal control methods are usually used to try to match the plan, not to plan directly. Usually, Supply Chain planners use a method such as the (s,Q) ordering policy we mention in the paper - it can be seen as a simplified PID. We expand further on why we do not use MPC in the requested changes (R11).
>
> W4: We hope that our response to R10 below helps to interpret the results.
>
> Second, as for the requested changes:
>
> -- R1: The actual demand is hidden from the agent, meaning that the full state of the model is indeed not observed.
>
> -- R2: We modeled the Q-values as a lambda distribution as it is flexible in terms of shape, prevents quantile crossing and results in less parameters to estimate. These points are missing. We will add them in Section 4.2 of the paper.
>
> -- R3: We agree that the name might cause some confusion. We will change it to T- DQN, referring to the original Tukey’s lambda distribution.
>
> -- R4: This is true, thanks for spotting this mistake. We will correct this in Table 1.
>
> -- R5: We use SELU because it shows good performance with no downside in theory (https://proceedings.neurips.cc/paper/2017/file/5d44ee6f2c3f71b73125876103c8f6c4-Paper.pdf). Moreover, in preliminary experiments we saw that it results in a slightly higher performance for DQN.
>
> -- R6: Figure 2 shows the non-normality of the runs, and explains why looking at a classic standard deviation leads to misinterpretation of the data. We will emphasize this in Figure 2.
>
> -- R7: You are right that it refers to Q-values. We will correct this mistake.
>
> -- R8: Good question! We did not notice this and fixed the summation sizes of Equation (1).
>
> -- R9: With perfect knowledge, aleatoric uncertainty would be inconsequential in terms of forecast. If one were to know the exact detail for all customers, there would be close to no randomness in the forecast. We will add this clarification to Section 3.1.3.
>
> -- R10: We relaunched our experiments. We will add learning curves to Section 5.
>
> -- R11: MPC takes a desired stock level as an input, and relies on a specified model to attain its target stock level. However, in our work, we aim to find the optimal stock level instead, without relying on a model. Thus, MPC can not be a meaningful baseline in this work, if we want the comparison to be fair. However, it is clear that using MPC would make sense downstream, after the model. Indeed, given a recommendation provided by an algorithm such as T-DQN (GLDQN), MPC would then be used on actual systems to try to match the recommendation.

---

> > ### Comment · Reviewer_w2jW · 2023-03-03
> > **Response to author rebuttal**
> >
> > Thanks for your responses. I will re-read the paper when you have made the edits. Two things in your response confused me and I am hoping you can clarify them:
> >
> > 1. Can you clarify why the environment is a POMDP rather than simply an MDP? What information about the state is missing from the observation?
> > -- R1: The actual demand is hidden from the agent, meaning that the full state of the model is indeed not observed.
> >
> > I still don't understand this. MDPs can have stochastic transitions, the agent can make an action and not know exactly what state it will end up in. Just because something is random doesn't mean it's a POMDP, the random demand can be modelled as a stochastic transition, eg, I have 10 loaves of bread and my action is to buy 5 more, then I stochastically transition to another state between {0, ..., 15} based on random purchase demand.
> >
> > 9. Why does ' forecast inaccuracy derive from epistemic uncertainty'? Surely there is aleatoric noise in the process being forecasted too.
> > -- R9: With perfect knowledge, aleatoric uncertainty would be inconsequential in terms of forecast. If one were to know the exact detail for all customers, there would be close to no randomness in the forecast. We will add this clarification to Section 3.1.3.
> >
> > Again I do not understand this response. For example, if I have a biased coin then each flip is the outcome of both epistemic uncertainty (what is the probability of heads) and aleatoric uncertainty (what is the outcome of the flip, given the probability of heads). If I have *perfect* knowledge of the probability of heads (ie, a perfect model) then I still have 'forecast' error in terms of the aleatoric uncertainty. I can tell you that the probability of heads is 75%, ie, my forecast is 0.75 heads from one flip, but you might still see a tails due to irreducible aleatoric noise.

---

> > > ### Author Response · Authors · 2023-03-14
> > > **Response to comment of Reviewer w2jW**
> > >
> > > Dear Reviewer w2jW,
> > >
> > > Thank you for following up on our answer!
> > >
> > > We hope that we can bring more clarity with this comment.
> > >
> > > 1) You are right that the stochastic demand does not fully describe a POMDP. We did not clarify that customers visiting in the morning will not come in the afternoon, and vice-versa. Indeed, our environment assumes a multivariate distribution on store visits with a negative correlation relative to the previous period. This means that the previous number of customers, which is not part of the state, is hidden from the agent. We will clarify this in the paper (Section 3.1.3).
> > >
> > > 2) With our answer, we meant to express that a model that would, for instance, know the current composition of a user’s fridge, the number of people in the household, as well as the user’s full browsing history, would drastically reduce the weight of aleatoric uncertainty. But “total surveillance” is not something we desire.
> > >
> > > Thank you & best regards,
> > >
> > > The authors

---

### Review · Reviewer_sE4U · 2023-03-12

**Summary Of Contributions:**

This paper study the problem of applying reinforcement learning for waste reduction. A RL environment RetailL is proposed, which includes the details of item generation, demand generation, forecast generation and stock update. Then a distributional rl algorithm , GLDQN is proposed , and outperforms the previous RL baselines in both reward and waste reduction cases.

**Audience:**

Yes

**Broader Impact Concerns:**

None.

**Claims And Evidence:**

Yes

**Requested Changes:**

See Weaknesses. Please address these points and clarify my questions.

**Strengths And Weaknesses:**

Strengths:
1.The RetaiL environments benefit the RL for real life community.
2.A new rl algorithm is proposed and outperforms previous baselines.

Weaknesses:
1.The POMDP formulation lacks more formal definition, such as the definition of the observation space.
2.What is the length of the episode?
3.How to evaluate the accuracy of the environment modeling?
4.How to choose the value for lambda_1,...,lambda_4 in Eq.(4)?
5.How many random seeds are used in experiments? In common, RL algorithms are evaluated in 5 random seeds.

---

> ### Author Response · Authors · 2023-03-14
> **Response to Reviewer sE4U**
>
>
> Dear Reviewer sE4U,
>
> We want to thank you for taking the time to read and assess our paper.
> To answer your questions in order:
>
> -- Q1: We will include further details on the POMDP in Section 3.1.3.
>
> -- Q2: Episodes have a length of 2000, to ensure agents are exposed to several cycles of waste.
>
> -- Q3: We built the environment using inputs from an actual grocery chain. While this does not mean the environment can represent just any grocery store, tunable parameters such as the number of deliveries per day, store frequentation and assortment should help in adapting the environment to other stores if need be.
>
> -- Q4: The lambda values in equation (4) are the output parameters of our neural network.
>
> -- Q5: We ran our experiments on 30 seeds.
>
> Best regards,
> The authors

---

### Author Response · Authors · 2023-03-14
**Response to reviewers**

Dear Reviewers,

We want to thank you all for taking the time to help us improve our paper. We feel that our current manuscript has been greatly improved, in both context and clarity thanks to your comments.

We are currently finishing the experiments that you requested and hope to share the revised paper with the additional results in a few days.

Thanks again & best regards,

The authors

---

> ### Author Response · Authors · 2023-03-25
> **Revision uploaded**
>
> Dear Reviewers,
>
> We want to thank you again for your comments. We have uploaded a revision of the paper that includes our response to your comments, as well as the requested changes.
>
> On top of highlighting the main changes, we added an Appendix containing training curves and code samples, added an experiment for an increased weight of waste, and added MAD (Median Absolute Deviation) to the score tables.
>
> Thanks again and best regards,
>
> The authors

---

### Decision · Action_Editors · 2023-04-20

**Recommendation:** Accept as is

**Comment:**

A RL environment RetailL is proposed, which includes the details of item generation, demand generation, forecast generation and stock update. Then a distributional rl algorithm , GLDQN is proposed , and outperforms the previous RL baselines in both reward and waste reduction cases. The authors carefully responded to all questions and all reviewers unanimously propose accept.

**Audience:**

Practical domain, useful for the RL audience.  Good for benchmarking (RL) algorithms to manage perishable inventory in a shop/warehouse. The goal in this domain is to maximize profits and minimize waste. Observations correspond to features of the product (cost price, selling price, shelf life, popularity) and also an approximate forecast of its demand. These features are derived from (anonymized) realistic data. Actions correspond to the number of new units to order. Items get replenished based on order and get consumed based on a consumption model.

**Claims And Evidence:**

Paper presents a new simulation environment for inventory management and proposes a new type of Q-learning algorithm to solve it.